# Antimicrobial Properties and Cytotoxic Effect of Imidazolium Geminis with Tunable Hydrophobicity

**DOI:** 10.3390/ijms222313148

**Published:** 2021-12-05

**Authors:** Syumbelya K. Amerkhanova, Alexandra D. Voloshina, Alla B. Mirgorodskaya, Anna P. Lyubina, Darya A. Kuznetsova, Rushana A. Kushnazarova, Vasilii A. Mikhailov, Lucia Ya. Zakharova

**Affiliations:** 1Arbuzov Institute of Organic and Physical Chemistry, FRC Kazan Scientific Center, Russian Academy of Sciences, 8 Arbuzov st., 420088 Kazan, Russia; syumbelya07@mail.ru (S.K.A.); microbi@iopc.ru (A.D.V.); mirgoralla@mail.ru (A.B.M.); aplyubina@gmail.com (A.P.L.); dashyna111@mail.ru (D.A.K.); ruwana1994@mail.ru (R.A.K.); 2L.M. Litvinenko Institute of Physical Organic Chemistry and Coal Chemistry, R. Luxemburg st., 70, 83114 Donetsk, Ukraine; v_mikhailov@yahoo.com

**Keywords:** dicationic imidazolium surfactants, antimicrobial activity, cytotoxicity, hemolysis, antimicrobial resistance, selectivity

## Abstract

Antimicrobial, membranotropic and cytotoxic properties of dicationic imidazolium surfactants of n-s-n (Im) series with variable length of alkyl group (n = 8, 10, 12, 14, 16) and spacer fragment (s = 2, 3, 4) were explored and compared with monocationic analogues. Their activity against a representative range of Gram-positive and Gram-negative bacteria, and also fungi, is characterized. The relationship between the biological activity and the structural features of these compounds is revealed, with the hydrophobicity emphasized as a key factor. Among dicationic surfactants, decyl derivatives showed highest antimicrobial effect, while for monocationic analogues, the maximum activity is observed in the case of tetradecyl tail. The leading compounds are 2–4 times higher in activity compared to reference antibiotics and prove effective against resistant strains. It has been shown that the antimicrobial effect is not associated with the destruction of the cell membrane, but is due to specific interactions of surfactants and cell components. Importantly, they show strong selectivity for microorganism cells while being of low harm to healthy human cells, with a SI ranging from 30 to 100.

## 1. Introduction

Antimicrobial resistance is a serious problem that hinders further progress in the fight against infectious diseases, and may also lead to their rapid spread in the future. The resistance of infectious agents to antimicrobial drugs complicates the process of their treatment and makes it less effective. Infectious diseases, in particular those caused by multi-resistant forms of microorganisms, currently pose a serious threat to human health. In this regard, there is a need to find alternative, highly effective and safe antimicrobial substances that will become the basis for new drugs. Surfactant molecules have unique features and a wide range of practical applications due to their amphiphilic nature. Their rich and diverse chemical composition confers a number of bioactive properties, including antioxidant, antiviral, anti-inflammatory, anti-cancer and anti-aging effects [1]. The antimicrobial properties of surfactants should be especially emphasized. Under natural conditions, microbial cells have a general negative charge, so cationic surfactants have found the most widespread practical use, which detrimentally affect Gram-positive and Gram-negative bacteria, yeast and mycelial fungi. Adsorption of surfactant on the surface of a microbial cell is the first stage of interaction of microorganisms with a chemical compound. The microelectrophoresis method shows that cationic surfactants adsorb on the cell surface of bacteria, reduce its charge, and in some cases can induce the recharging of cell wall [2]. In addition, the plasmid loss in bacterial cells may occur as well. Probably, it is the impaired permeability that explains the increased sensitivity of resistant strains of bacteria to antibiotics in the presence of surfactants. The end result of the surfactant action on the microbial cell is the destruction of the cell wall [3,4].

The nature and charge of the polar head group and alkyl tail length are the key structural characteristics of surfactants responsible for their basic biological properties. Imidazolium cationic surfactants attract much attention in the chemotherapy due to their high antimicrobial activity in combination with allowable toxic properties [5,6,7,8,9]. Recently, dicationic gemini surfactants have been actively used in pharmacology and biotechnology. This is wide class of compounds bearing two hydrophobic tails and two positively charged head groups covalently bound by a spacer moiety. Geminis demonstrate beneficial aggregation properties and functional activity over their monocationic analogues [10,11,12,13]. In review [10], a marked decrease in critical micellar concentrations (CMC) and superior surface activity of geminis are emphasized compared to monomeric counterparts. Their aggregation and morphological behavior are discussed, with both fundamental aspects and practical applications highlighted. Self-assembly of dimeric and oligomeric surfactants is reviewed in [11]. Apart from single gemini solutions, their mixed systems with polymers and conventional surfactants are covered as well. The authors of [13] focus on enhanced solubilization capacity of geminis toward spectral probe thymolphthalein and anti-inflammatory drug indomethacin, which is probably due to multi-centered interactions of surfactant aggregates with incapsulated substrates. Importantly, higher antimicrobial activity of gemini surfactants is observed in some cases [14,15,16]. It is worth noting that dicationic surfactants with imidazolium head group have recently been reported: the variation of alkyl tail length and distance between cationic centers [17,18,19], introduction of substituents into the spacer fragment [20], as well as modification of the structure of the imidazolium head group [21] allows for controlling their aggregation behavior and properties, including antimicrobial activity. Previously, the biological activity of dicationic imidazolium surfactants with decyl and dodecyl tails have been studied in our research group, which showed high antimicrobial effects against a number of test microorganisms, with low cytotoxic activity observed [22]. In the present work, systematic study of antimicrobial, membranotropic and toxic properties is carried out for the representative series of imidazolium geminis, including lower and higher homologues, as well as for their monocationic analogues, which allows for identifying the relationship between biological activity and structural features of amphiphilic compounds. The formulas of the imidazolium surfactants are given below (Figure 1).

In addition, it was supposed to expand the range of tested microorganisms, to include resistant strains. The identified structure–property correlations can serve as the basis for the directed design of bioactive surface-active compounds used in pharmacology and medicine.

## 2. Results and Discussion

### 2.1. Antimicrobial Activity

Synthesized dicationic (n-s-n (Im)) and monocationic (Im-n) imidazolium surfactants were tested on antimicrobic activity in relation to a number of Gram-positive bacteria of *Staphylococcus aureus ATCC 6538P FDA 209P* (*Sa*), *Bacillus cereus ATCC 10702* (*Bc*), *Enterococcus faecalis ATCC 29212* (*Ef*), against methicillin-resistant strains of *Staphylococcus aureus*—*MRSA-1* and *MRSA-2* and Gram-negative bacteria of *Escherichia coli ATCC* 25922. Strains *MRSA-1* and *MRSA-2* demonstrate high level of drug resistance. *MRSA-1* has developed resistance to antibiotics of the fluoroquinolone and β-lactam series, while *MRSA-2* only to β-lactams). Antifungal activity has been studied against *Candida albicans ATCC 10231* and *Trichophyton mentagrophytes var. gypseum 1773*. Table 1 summarizes the previously published and new data on antimicrobial activity of the studied dicationic imidazolium surfactants. 

It follows that the length of the hydrophobic tail is the main structural factor responsible for the effectiveness of the test compounds, while the number of methylene units in the spacer fragment has significantly lower effect. Decyl derivatives are found to be leading compounds; they exhibit a wide spectrum of antimicrobial activity, similar or even superior compared to known drugs.

Slightly less activity is shown by dodecyl derivatives. The compounds n-s-n (Im) for which n = 8 or 14 appeared significantly less effective, and their analogues with hexadecyl tail have no antibacterial and antifungal activity against the entire range of test microorganisms. It should be noted that for gemini surfactants with tetradecyl tails, the dependence of antimicrobial activity on the spacer length occurred. The compounds lost antimicrobial activity, when the spacer fragment increased to three and four methylene groups.

Unlike gemini surfactants, none of their monocationic analogues are characterized by such high activity (Table 2, Figure 2). The antimicrobial effect of the compounds of the Im-n series increases with growing of their lipophilicity and, depending on the test object, reaches a maximum for derivatives with tetra- or hexadecyl tails. The dependence between the minimum inhibitory concentration (MIC) and the number of carbon atoms in the alkyl substituent (Figure 2) clearly demonstrates the different influence of hydrocarbon chain length in mono- and dicationic imidazolium surfactants on their antimicrobial effects.

It should be noted that the compounds of the series n-s-n (Im) show significant activity not only against Gram-positive, but also against Gram-negative bacteria. The latter have an additional outer membrane preventing access to the cell of foreign membranotropic compounds and increasing its resistance to antimicrobial agents, including surfactants [22,23]. However, in relation to *Escherichia coli*, compounds 10-s-10 (Im) are effective at the level of reference drugs (Table 1). This result encouraged us to test dicationic imidazolium surfactants toward other antibiotic-resistant strains as well.

The effects of these compounds on the pathogen of a wide range of in-hospital human infections, *Enterococcus faecalis*, were evaluated. The clinical significance of the genus *Enterococcus* is directly related to its antibiotic resistance. *Enterococcus* resistance to beta-lactam antibiotics, low doses of aminoglycosides, vancomycin, cephalosporins, fluoroquinolones [24,25,26] is described. In the present work, new data have been obtained indicating that the imidazolium surfactants studied inhibit the viability of this pathogen.

The highest antimicrobial activity against *Enterococcus faecalis* was shown by dicationic surfactants bearing a decyl and dodecyl hydrophobic tail, which were 2–4 times higher in activity than the antibiotic of the fluoroquinolone series norfloxacin (Table 1). Importantly, gemini imidazolium surfactants were also active against methicillin-resistant strains of *S. aureus*—*MRSA-1* and *MRSA-2* which pose a serious threat to public health [27,28]. Derivatives with decyl and dodecyl tails appeared to be leader compounds for resistant strains of *S. aureus*, while compounds with octyl or tetradecyl substituents showed only moderate antimicrobial activity (Table 1). It should be noted that the tested dicationic imidazolium surfactants demonstrated activity not only to bacteria, but also to fungi (*Trichophyton mentagrophytes var. gypseum* and *Candida albicans*), which is pronounced for compounds with decyl and dodecyl tails.

### 2.2. Crystal Violet Assay

When studying the mechanism of antimicrobial action of surfactants, researchers investigate their adsorption and the formation of complexes on the surface of a microbial cell, change of microbial cell permeability under the influence of these compounds and determine their actions on physiological processes and enzymatic activity of microorganisms. According to current concepts, the cell wall and cytoplasmic membrane of bacteria are targeted for many antimicrobial agents, including surfactants [29,30,31]. It is known that chemicals can affect microorganisms specifically and non-specifically. The specific mechanism manifests at very low concentrations of the antimicrobial compound, which can react with certain components of the cell, disrupting their normal functioning by inhibition of some physiological processes and enzymatic activity of microorganisms [32,33]. In particular, there are reports of stimulation of respiratory activity and some enzymes in the presence of surfactants [34,35]. The researchers suggest that the enzyme systems of cells are damaged secondarily, while the surface and internal membrane structures associated with many enzymes are changed primarily. At the same time, a short stimulation of the activity of enzymes dissolved in the liquid part of the cytoplasm can be observed with the cell, seeming to replace the damaged enzyme system with the intact system [36].

A nonspecific effect on the cell usually manifests at sufficiently high concentrations of substances. It may be associated with an adverse change in surface tension, pH, high osmotic pressure and so on. In the literature, the specific effect of surfactants on microorganisms is described [1,37,38].

Based on the literature data, we evaluate the change in the permeability of *S. aureus* cells under the influence of cationic geminis. To do this, a hydrophobic crystal violet (CV) dye was used, which poorly penetrates the membrane of viable cells, but easily passes when they are damaged. The ability to absorb the dye from the medium is an important marker of the state of cytoplasmic membrane [39]. Studies were conducted in a concentration range that included MIC and MBC of compounds n-3-n (Im) at n = 8, 10, 12 and 14 (Table 1). Figure 3 shows that compound 8-3-8 (Im) has a very weak membranotropic effect. Even at the highest concentration (150 μM), exceeding its MBC (28.5 μM), the CV uptake was only 5%. Treatment of *S. aureus* cells with leader compounds (10-3-10 (Im) and 12-3-12 (Im)) in the range of their MIC and MBC (0.8–11 μM) CV uptake did not exceed 3%. An increase in the lipophilicity of the gemini surfactants to tetradecyl derivative led to an increase in membranotropic properties. Compound 14-3-14 (Im) in the MBC range (75–150 μM) was shown to have a significant effect on the permeability of the *S. aureus* membrane, increasing CV uptake to 40–63%.

The results suggest that the antimicrobial effects of compounds with an octyl, decyl and dodecyl tails are primarily related to their specific effects on microorganisms. The low concentrations corresponding to the MIC and MBC of these compounds are significantly lower than those corresponding to a violation of the integrity of the cytoplasmic membranes. This suggests that surfactants interact with certain components of the cell, impeding their normal functioning. Moreover, the presence of a decyl tail in the surfactant molecule leads to the most outstanding results, namely, the compounds acting bactericidally in very low concentrations against sensitive and resistant strains of *S. aureus*. With an increase in the length of the lipophilic substituent to tetradecyl, the gemini surfactants probably lose the ability to specifically bind to the surface of the bacterial cell.

### 2.3. Incorporation of Dicationic Imidazolium Surfactants into Lipid Bilayers Mimicking Biomembranes

The hypothesis regarding the specific mechanism of action of gemini surfactants is consistent with the turbidimetry data on the incorporation of imidazolium surfactants into lipid bilayers. The observed change in the temperature of the main phase transition of the lipid indicates the integration of surfactant into the lipid bilayer, which leads to its disordering, loosening, and disruption of permeability [40,41]. In this experiment, the dipalmitoylphosphatidylcholine (DPPC)-based liposome dispersion was titrated with a solution of gemini imidazolium surfactants with different hydrocarbon tail lengths. Figure 4 shows the dependence of the main phase transition temperature on the surfactant/DPPC molar ratio for the homologous series of gemini imidazolium surfactants. The content of gemini surfactants in the test samples corresponded to the range of their concentrations in the CV uptake experiment. The slope of the initial region of dependence reflects the degree of influence of the surfactant on the properties of the membrane, which varied as follows: 8-3-8 (Im) < 10-3-10 (Im) < 12-3-12 (Im). The surfactant’s effect on the membrane properties enhances with an increase in concentration, and at some critical surfactant/lipid ratio, the bilayer structure breaks down and the membrane loses its properties. This transition is reflected in the plot by a kink of a curve (Figure 4). Antimicrobial properties for the tested surfactants manifested at significantly lower concentrations, which suggests that their activity is not associated with complete destruction of the lipid bilayer, but is caused by specific interactions. This is supported by the fact that tetradecyl surfactant destroys bilayer more easily than other analogs (at a significantly lower concentration of surfactant); however, it does not show pronounced antimicrobial properties.

### 2.4. Hemolytic and Cytotoxic Activity of the Test Compounds

An important characteristic in assessing the biological activity of new chemical compounds is their cytotoxic effect on mammalian cells. The ability of the test compound to cause destruction of human blood erythrocytes shows its toxic effect on the internal environment of the body. In vitro hemolysis assay is a simple screening test that may help in studying cytotoxicity on more complex models [42]. Dicationic and monocationic imidazolium surfactants were tested for cytotoxicity against blood erythrocytes and human hepatocyte cell line Chang liver (Figure 5 and Figure 6). Data on hemolytic and cytotoxic activity are presented in terms of HC_50_ (concentration of the test compound that causes 50% erythrocyte hemolysis) and IC_50_ (concentration of the test compound that causes 50% cell death of the cell population).

It can be seen that imidazolium geminis (especially leader compounds of the 10-s-10 (Im) series) do not exhibit high cytotoxic and hemolytic activity. An insignificant increase is observed only in gemini surfactants of the 12-s-12 (Im) series. In contrast to gemini imidazolium surfactants, the cytotoxic effect of their monocationic analogs increases with the growth of the alkyl chain and reaches the maximum value for hexa- and octadecyl homologs. It should be noted that all tested compounds have lower toxicity to human erythrocytes and hepatocytes compared to the typical cationic surfactant cetyltrimethylammonium bromide (CTAB).

The selectivity of compounds to microbial cells is an important criterion for evaluating cytotoxic action. This parameter is characterized by a selectivity index (SI) value, which is calculated as the ratio between the HC_50_ value for erythrocytes (IC_50_ for eukaryotic cells) and the MIC value for microbial cells. It can be seen that the highest selectivity to the *S. aureus 209* strain is demonstrated by imidazolium surfactants with decyl tails, which is coupled with low toxicity to eukaryotic cells of human blood and liver with (Table 3). The obtained data characterize the safety of the gemini surfactants in comparison with their monocationic analogues against healthy human cells, which is a key criterion for the development of potential antimicrobial agents.

## 3. Materials and Methods 

### 3.1. Chemistry

Monocationic imidazolium surfactants were synthesized by the interaction of methylimidazole with the corresponding halide alkyl, in analogy with [43,44]. Dicationic imidazolium salts with long-chain alkyl tails and polymethylene bridges (spacer) have been synthesized in two stages in accordance with [22]: in the first step, imidazole was mono-alkylated, and in the second step, the bridge was introduced (Figure 1).

The structure of the synthesized compounds was confirmed by elemental analysis, IR and NMR spectroscopy. The obtained characteristics corresponded to the literature data [22,44]. As the example, the characteristics of bis-1,4-(3′-dodecylimidazolium-1′-yl)butane dibromide, 12-4-12 (Im) are given below. 1H NMR (400 MHz, DMSO-d_6_, δ, ppm): 9.19 (s, 2H), 7.79 (br s, 2H), 7.77 (br s, 2H), 4.18 (m, 4H), 4.13 (tr, 4H), 1.82-1.72 (m, 8H), 1.29-1.16 (m, 36H), 0.83 (tr, J = 6.5 Hz, 6H). Anal. Calcd for C_34_H_64_Br_2_N_4_: C, 59.29; H, 9.37; N, 8.14; Br 23.20. Found: C, 58.99; H, 9.59; N, 8.05; Br 23.01. IR (KBr): 3400br, 2910, 2840, 1560, 1470, 1165, 820, 620.

Commercially available DPPC, CV, 4′,6-diamidino-2-phenylindole (DAPI) and propidium iodide (Sigma-Aldrich, Schnelldorf, Germany) were used without further purification. 

### 3.2. Antimicrobial Activity

The antimicrobial activity of the test compounds was determined by serial dilutions in Muller–Hinton broth for bacterial cultivation and in Sabouraud dextrose broth for fungal pathogens according to known procedures [22]. Gram-positive bacteria cultures were used for the experiment: *Staphylococcus aureus ATCC 6538 P FDA 209P*, *Bacillus cereus ATCC 10702 NCTC 8035*, *Enterococcus faecalis ATCC 29212*; Gram-negative bacteria: *Escherichia coli ATCC 25922, Pseudomonas aeruginosa ATCC 9027* and fungi: *Candida albicans ATCC 10231* and *Trichophyton mentagrophytes var. gypseum 1773*. Methicillin-resistant strains of *S. aureus* (*MRSA*) were isolated from the body of patients with chronic tonsillitis (*MRSA-1*) and sinusitis (*MRSA-2*) in the bacteriological laboratory of the Republican Clinical Hospital (Kazan, Russia). The bacterial load was 3.0 × 10^5^ cfu/mL. The results were recorded every 24 h for 5 days. Cultures were incubated at 37 °C. The experiment was repeated three times. The dilutions of the compounds were prepared immediately in nutrient media. The MIC was defined as the minimum concentration of a compound that inhibits the growth of the corresponding test microorganism. The growth of bacteria as well as the absence of the growth due to the bacteriostatic action of imidazolium surfactant were recorded. To determine MBC, an aliquot of the bacterial culture was transferred onto Mueller–Hinton agar in a 10 cm Petri dish and incubated for 24 h at 37 °C. MBC was the minimal concentration of the tested compound at which bacterial colonies were not detected, indicating that the bacteria were killed with an efficiency of >99.9% The experiments were performed in triplicate.

### 3.3. Hemolytic Activity

The hemolytic activity of the imidazolium surfactants was evaluated according to the known method [22,45] by comparing the optical density of the solution holding the test compound with the optical density of the human blood solution at 100% hemolysis measured by the microplate reader Invitrologic (Russia) at λ = 540 nm. A control sample corresponding to zero hemolysis (negative control) was prepared by adding 10% erythrocyte suspension to saline (0.9% NaCl). A control sample corresponding to 100% hemolysis was prepared by adding 10% erythrocyte suspension to distilled water (positive control).

### 3.4. CV Assay

The change in the absorption of the CV dye by *Staphylococcus aureus* cells grown on Muller–Hinton agar was determined as described in [45]. A night culture of *S. aureus 209P* was cultured on the day of the experiment in Muller–Hinton broth until the culture reached the middle of the exponential growth phase and then centrifuged. The precipitate was washed twice with 0.01 M phosphate-buffered solution (PB) followed by centrifugation at 2152× *g* for 10 min. The washed cells were resuspended in the PB to obtain 2 × 10^8^ CFU/mL. The resulting inoculum was added to the test compounds at a ratio of 1:1 and incubated for 30 min at 37 °C. CV dye was then added to the solution, the concentration of which was 0.001% in the run. The solution was incubated for 10 min, then centrifuged at 12,396× *g* for 2 min. The optical density (*OD*) of the supernatant was determined at a wavelength of 540 nm on the microplate reader Invitrologic (Russia). The percentage of CV uptake was calculated by the formula:$$CV=\frac{OD_{\mathrm{sample}}-OD{\;}_{\mathrm{control}}}{OD_{\mathrm{CV}\;\mathrm{solution}}}\times 100,$$
where *OD*_sample_, *OD*_control_ and *OD*_solution_ is the value of optical density of the sample, of the control (without test-compound) and of the CV solutions (without cells), respectively.

### 3.5. Turbidimetric Measurements

DPPC liposomes for turbidimetric measurements were prepared as follows: 5.4 mg of DPPC was dissolved in 60 μL CHCl_3_ and kept overnight at room temperature for solvent evaporation. Obtained lipid thin film was suspended in water at 60 °C and vigorous stirring. After that, the dispersion was heat-treated: alternately cooled down in liquid nitrogen and then heated in water bath at 60 °C. This procedure was repeated three times. Suspension was extruded through LiposoFast Basic («Avestin» Ottawa, ON, Canada) extruder using carbohydrate filter with 100 nm pore diameter. Prepared liposomes were diluted to 0.7 mM and used in turbidimetric measurements at Specord 250 PLUS («Analytik Jena», Jena, Germany) spectrophotometer. Primary turbidimetric plots were treated in terms of Van’t Hoff two-state model. In accordance with this approach, the break point in the plot corresponds to the main phase transition of DPPC from gel to liquid crystalline phase [46].

### 3.6. Cytotoxicity Assay 

The cytotoxic activity of the studied compounds against a healthy human hepatocyte cell line (Chang liver) was evaluated using the multifunctional Cytell cell imaging system (GE Health Care Life Science, Uppsala, Sweden) using the Cell Viability Bio application, which accurately counts the number of cells and evaluates their viability based on the intensity of fluorescence [45]. Two fluorescent dyes (DAPI and Propidium iodide) were used in the experiments. DAPI is able to penetrate intact membranes of living cells and colors the nuclei in blue. Propidium iodide dye pervades only dead cells with damaged membranes, staining them in yellow. The work used the human hepatocyte cell line Chang liver from Gamaleya Research Institute of Epidemiology and Microbiology. Calculation of IC_50_ was made by means of the online calculator: MLA—“Quest Graph Calculator ™ IC50”. AAT Bioquest, Inc, 25 July 2019, https://www.aatbio.com/tools/ic50-calculator (accessed on 3 February 2021). The experiments were repeated three times. Intact cells cultured in parallel with experimental cells were used as a control. 

## 4. Conclusions

In this work, the antimicrobial, membranotropic, and cytotoxic properties of the n-s-n (Im) series of surfactants with a variable length of alkyl tail (n = 8, 10, 12, 14, 16) and the spacer fragment (s = 2, 3, 4) were studied, with the relation between the structural features and biological activity elucidated. Dicationic surfactants with decyl tail demonstrated the highest antimicrobial effect, while the maximum activity for monocationic analogues is observed for tetradecyl derivatives. The leading compounds are 2–4 times higher in activity compared to reference antibiotics and prove effective against resistant strains. For example, in the case of *Enterococcus faecalis* which is the pathogen of a wide range of hospital acquired human infections infections with significant resistance to fluoroquinolones and beta-lactam antibiotics, MIC is 5.8 µM for 12-2-12 (Im), while for norfloxacin this value is 24 µM. Analysis of data on cell permeability obtained with the dye CV, as well as the study of incorporation of dicationic imidazolium compound into lipid bilayers made it possible to assume that the antimicrobial effect is not associated with the destruction of the cell membrane, but is due to specific interactions of surfactants and cell components. Thus, for leader decyl and dodecyl derivatives, very low CV uptake of 3% occurs in the range of their MIC and MBC. The cytotoxic effect of monocationic imidazolium surfactants increases with the growth of the alkyl chain and reaches the maximum value for hexa- and octadecyl homologs. In contrast to this trend, gemini imidazolium surfactants do not exhibit high cytotoxic and hemolytic activity. It is noteworthy that imidazolium geminis showed lower toxicity to human erythrocytes and hepatocytes compared to the typical cationic surfactant CTAB. In addition, they have high selectivity for microorganism cells and are low toxic on normal human cells with a selectivity index SI (HC_50_/MIC) exceeding 100 and SI (IC_50_/MIC) exceeding 30. The results nominate the surfactants studied as a basis for targeted design of bioactive amphiphilic compounds for the development of new antimicrobial agents. 

## Data Availability

The data presented in this study are available on request from the corresponding author: Zakharova Lucia.

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
