# Peer review of "Antimicrobial Properties and Cytotoxic Effect of Imidazolium Geminis with Tunable Hydrophobicity"

_ijms, 2021, doi:10.3390/ijms222313148_

Round 1
Reviewer 1 Report
The paper is interesting both in its research topic and in the research methodology applied. The conclusions seem to be promising. In general, the article is well prepared, but I suggest some minor revisions, i.e.:
1) Notation of references such as [8-13] or [14-20] is not adequate. Authors should divide these ranges into smaller ones and discuss in more detail the content of the cited articles.
2) Section 4.1.: synthesis methodology of surfactants as well as their properties should be discussed briefly.
3) Section 4.2.: the procedure of antimicrobial activity assessment should be described briefly. The same applies to the procedure of CV assay - section 4.4.
4) Notation of reference 26- the whole journal name should be replaced by its abbreviation.
5) Final conclusions should be more numerical, additionally the font size in this section should be uniform.
Author Response
1) Notation of references such as [8-13] or [14-20] is not adequate. Authors should divide these ranges into smaller ones and discuss in more detail the content of the cited articles.
Reply: Thanks for your suggestion. We divided ranges of references [8-13] and [14-20] into smaller ones and discussed them in more detail (pages 2-3).
2) Section 4.1.: synthesis methodology of surfactants as well as their properties should be discussed briefly.
Reply: Thank you, in accordance with your comment synthesis methodology of surfactants was discussed (pages 12-13).
3) Section 4.2.: the procedure of antimicrobial activity assessment should be described briefly. The same applies to the procedure of CV assay - section 4.4.
Reply: Thank you very much. We added a description of the experiment to the text of the manuscript (pages 13-14).
4) Notation of reference 26- the whole journal name should be replaced by its abbreviation.
Reply: We replaced whole journal name by its abbreviation.
5) Final conclusions should be more numerical, additionally the font size in this section should be uniform.
Reply: Thanks for your suggestion. The final conclusions have been revised; the font size have been corrected (pages 11-12).
Reviewer 2 Report
The manuscript titled ‘’Antimicrobial Properties and Cytotoxic Effect of Imidazolium Geminis with Tunable Hydrophobicity’’ has been reviewed
The antibacterial, membranotropic, and cytotoxic features of dicationic imidazolium surfactants with varying alkyl chain and spacer length are investigated in this paper. The manuscript's scientific quality is very good, and a lot of work has been put in. The English language (especially in the introduction) has to be improved; some sentences are difficult to follow.
Line 21, rephrase the following sentence: s. Importantly, they demonstrate high selectivity for microorganism cells and are low toxic toward healthy 21 human cells with a selectivity index SI ranged from 30 to 100.
Suggest the following: Importantly, they show strong selectivity for microorganism cells while being low harmful to healthy human cells, with a SI ranging from 30 to 100.
Line 31: rephrase the following sentence: The amphiphilic nature of surfactant molecules causes (accounts) for their unique properties and wide practical applications.
Suggestion: Surfactant molecules have unique features and a wide range of practical applications due to their amphiphilic nature.
Line 39: the following evidences (findings) should be referenced: The microelectrophoresis method shows that cationic surfactants adsorb on the cell surface of 39 bacteria, reduce its charge, and in some cases can induce the recharging of cell wall. Interaction of surfactants with microbial cells is accompanied by changes of their properties, with the cell dimension and surface curvature transformed. Surfactants bind to components of the cytoplasmic membrane (CPM) and disrupt its normal functioning, thereby modifying the permeability and release profile of low-molecular metabolites from cells into the environment. I
Line 46: Did you mean destruction not distraction?
Figure 1: delete Russian alphabet (gde)
Line 69: ......including attracting resistant strains. Could be rephrased
Suggestion: to include resistant strains
Line 75: Rephrase the following sentence: Synthesized dicationic and monocationic imidazolium surfactants were tested on antimicrobic activity 75 in relation to a number of gram-positive bacteria….
Suggestion: Antimicrobial activity of the synthesized dicationic (Number) and monocationic (number) imidazolium surfactants was tested against a variety of gram-positive bacteria…….
Line 92: rephrase the following sentence; The leading compounds were decyl derivatives exhibiting a wide range of antimicrobial activity comparable with known drugs, and on some strains even exceed it.
Line 122: rephrase the following sentence: In the present work new data indicated
that tested imidazolium surfactants inhibited the viability of this pathogen were obtained.
Author Response
Line 21: Importantly, they demonstrate high selectivity for microorganism cells and are low toxic toward healthy 21 human cells with a selectivity index SI ranged from 30 to 100.
Suggest the following: Importantly, they show strong selectivity for microorganism cells while being low harmful to healthy human cells, with a SI ranging from 30 to 100.
Reply: Thank you very much for your suggestion. The sentence has been revised.
Line 31: rephrase the following sentence: The amphiphilic nature of surfactant molecules causes (accounts) for their unique properties and wide practical applications.
Suggestion: Surfactant molecules have unique features and a wide range of practical applications due to their amphiphilic nature.
Reply: Thank you very much for your suggestion. The sentence has been revised.
Line 39: the following evidences (findings) should be referenced: The microelectrophoresis method shows that cationic surfactants adsorb on the cell surface of 39 bacteria, reduce its charge, and in some cases can induce the recharging of cell wall. Interaction of surfactants with microbial cells is accompanied by changes of their properties, with the cell dimension and surface curvature transformed. Surfactants bind to components of the cytoplasmic membrane (CPM) and disrupt its normal functioning, thereby modifying the permeability and release profile of low-molecular metabolites from cells into the environment. I
Reply: Thank you very much for your suggestion. The reference has been added (refs.2-4).
Line 46: Did you mean destruction not distraction?
Reply: Thank you, typo-errors are removed in accordance with your remark.
Figure 1: delete Russian alphabet (gde)
Reply: Thank you, Russian alphabet has been deleted.
Line 69: ......including attracting resistant strains. Could be rephrased
Suggestion: to include resistant strains
Reply: Thank you for your suggestion. The sentence has been revised.
Line 75: Rephrase the following sentence: Synthesized dicationic and monocationic imidazolium surfactants were tested on antimicrobic activity 75 in relation to a number of gram-positive bacteria….
Suggestion: Antimicrobial activity of the synthesized dicationic (Number) and monocationic (number) imidazolium surfactants was tested against a variety of gram-positive bacteria…….
Reply: Thank you very much for your suggestion. The sentence has been rephrased.
Line 92: rephrase the following sentence; The leading compounds were decyl derivatives exhibiting a wide range of antimicrobial activity comparable with known drugs, and on some strains even exceed it.
Reply: Thank you very much for your suggestion. The sentence has been changed.
Line 122: rephrase the following sentence: In the present work new data indicated that tested imidazolium surfactants inhibited the viability of this pathogen were obtained.
Reply: Thank you very much for your suggestion. The sentence has been rephrased.
Reviewer 3 Report
I have no comments
Author Response
I have no comments -
Reply: Thank you very much.